# A Unified Semantic Embedding: Relating Taxonomies and Attributes

**Sung Ju Hwang**[*]
Disney Research
Pittsburgh, PA
sungju.hwang@disneyresearch.com

**Leonid Sigal**
Disney Research
Pittsburgh, PA
lsigal@disneyresearch.com

## Abstract

We propose a method that learns a discriminative yet semantic space for object categorization, where we also embed auxiliary semantic entities such as supercategories and attributes. Contrary to prior work, which only utilized them as side information, we explicitly embed these semantic entities into the same space where we embed categories, which enables us to represent a category as their linear combination. By exploiting such a unified model for semantics, we enforce each category to be generated as a supercategory + a sparse combination of attributes, with an additional exclusive regularization to learn discriminative composition. The proposed reconstructive regularization guides the discriminative learning process to learn a model with better generalization. This model also generates compact semantic description of each category, which enhances interoperability and enables humans to analyze what has been learned.

## 1 Introduction

Object categorization is a challenging problem that requires drawing boundaries between groups of objects in a seemingly continuous space. Semantic approaches have gained a lot of attention recently as object categorization became more focused on large-scale and fine-grained recognition tasks and datasets. Attributes [1, 2, 3, 4] and semantic taxonomies [5, 6, 7, 8] are two popular semantic sources which impose certain relations between the category models, including a more recently introduced analogies [9] that induce even higher-order relations between them. While many techniques have been introduced to utilize each of the individual semantic sources for object categorization, no unified model has been proposed to relate them.

We propose a unified semantic model where we can learn to place categories, supercategories, and attributes as points (or vectors) in a hypothetical common semantic space, and taxonomies provide specific topological relationships between these semantic entities. Further, we propose a discriminative learning framework, based on dictionary learning and large margin embedding, to learn each of these semantic entities to be well separated and pseudo-orthogonal, such that we can use them to improve visual recognition tasks such as category or attribute recognition.

However, having semantic entities embedded into a common space is not enough to utilize the vast number of relations that exist between the semantic entities. Thus, we impose a graph-based regularization between the semantic embeddings, such that each semantic embedding is regularized by sparse combination of auxiliary semantic embeddings. This additional requirement imposed on the discriminative learning model would guide the learning such that we obtain not just the optimal model for class discrimination, but to learn a semantically plausible model which has a potential to be more robust and human-interpretable; we call this model Unified Semantic Embedding (USE).

---

[*]Now at Ulsan National Institute of Science and Technology in Ulsan, South Korea

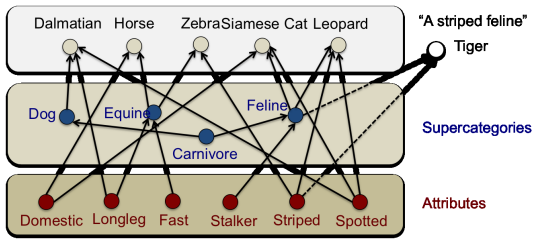

Figure 1: **Concept:** We regularize each category to be represented by its supercategory + a sparse combination of attributes, where the regularization parameters are learned. The resulting embedding model improves the generalization ability by the specific relations between the semantic entities, and also is able to compactly represent a novel category in this manner. For example, given a novel category *tiger*, our model can describe it as a *striped feline*.

The observation we make to draw the relation between the categories and attributes, is that a category can be represented as the sum of its supercategory + the category-specific modifier, which in many cases can be represented by a combination of attributes. Further, we want the representation to be compact. Instead of describing a dalmatian as a domestic animal with a lean body, four legs, a long tail, and spots, it is more efficient to say it is a spotted dog (Figure 1). It is also more exact since the higher-level category dog contains all general properties of different dog breeds, including indescribable dog-specific properties, such as the shape of the head, and its posture.

This exemplifies how a human would describe an object, to efficiently communicate and understand the concept. Such decomposition of a category into attributes+supercategory can hold for categories at any level. For example, supercategory feline can be described as a stalking carnivore.

With the addition of this new generative objective, our goal is to learn a discriminative model that can be compactly represented as a combination of semantic entities, which helps learn a model that is semantically more reasonable. We want to balance between these two discriminative and generative objectives when learning a model for each object category. For object categories that have scarce training examples, we can put more weight on the generative part of the model.

**Contributions:** Our contributions are threefold: (1) We show a multitask learning formulation for object categorization that learns a unified semantic space for supercategories and attributes, while drawing relations between them. (2) We propose a novel sparse-coding based regularization that enforces the object category representation to be reconstructed as the sum of a supercategory and a sparse combination of attributes. (3) We show from the experiments that the generative learning with the sparse-coding based regularization helps improve object categorization performance, especially in the one or few-shot learning case, by generating semantically plausible predictions.

## 2   Related Work

**Semantic methods for object recognition.**   For many years, vision researchers have sought to exploit external semantic knowledge about the object to incorporate semantics into learning of the model. Taxonomies, or class hierarchies were the first to be explored by vision researchers [5, 6], and were mostly used to efficiently rule out irrelevant category hypotheses leveraging class hierarchical structure [8, 10]. Attributes are visual or semantic properties of an object that are common across multiple categories, mostly regarded as describable mid-level representations. They have been used to directly infer categories [1, 2], or as additional supervision to aid the main categorization problem in the multitask learning framework [3]. While many methods have been proposed to leverage either of these two popular types of semantic knowledge, little work has been done to relate the two, which our paper aims to address.

**Discriminative embedding for object categorization.**   Since the conventional kernel-based multiclass SVM does not scale due to its memory and computational requirements for today's large-scale classification tasks, embedding-based methods have gained recent popularity. Embedding-based methods perform classification on a low dimensional *shared* space optimized for class discrimination. Most methods learn two linear projections, for data instances and class labels, to a common lower-dimensional space optimized by ranking loss. Bengio et al. [10] solves the problem using stochastic gradient, and also provides a way to learn a tree structure which enables one to efficiently predict the class label at the test time. Mensink et al. [11] eliminated the need of class embedding by replacing them with the class mean, which enabled generalization to new classes at near zero cost.

There are also efforts in incorporating semantic information into the learned embedding space. Weinberger et al. [7] used the taxonomies to preserve the inter-class similarities in the learned space,

in terms of distance. Akata et al. [4] used attributes and taxonomy information as labels, replacing the conventional unit-vector based class representation with more structured labels to improve on zero-shot performance. One most recent work in this direction is DEVISE [12], which learns embeddings that maximize the ranking loss, as an additional layer on top of the deep network for both images and labels. However, these models impose structure only on the output space, and structure on the learned space is not explicitly enforced, which is our goal.

Recently, Hwang et al. [9] introduced one such model, which regularizes the category quadruplets, that form an analogy, to form a parallelogram. Our goal is similar, but we explore a more general compositional relationship, which we learn without any manual supervision.

**Multitask learning.** Our work can be viewed as a multitask learning method, since we relate each model for different semantic entities by learning both the joint semantic space and enforcing geometric constraints between them. Perhaps the most similar work is [13], where the parameter of each model is regularized while fixing the parameter for its parent-level models. We use similar strategy but instead of enforcing sharing between the models, we simply learn each model to be close to its approximation obtained using higher-level (more abstract) concepts.

**Sparse coding.** Our method to approximate each category embedding as a sum of its direct super-category plus a sparse combination of attributes, is similar to the objective of sparse coding. One work that is specifically relevant to ours is Mairal et al. [14], where the learning objective is to reduce both the classification and reconstruction error, given class labels. In our model, however, the dictionary atoms are also discriminatively learned with supervision, and are assembled to be a semantically meaningful combination of a supercategory + attributes, while [14] learns the dictionary atoms in an unsupervised way.

## 3 Approach

We now explain our unified semantic embedding model, which learns a discriminative common low-dimensional space to embed both the images and semantic concepts including object categories, while enforcing relationships between them using semantic reconstruction.

Suppose that we have a $d$-dimensional image descriptor and $m$-dimensional vector describing labels associated with the instances, including category labels at different semantic granularities and attributes. Our goal then is to embed both images and the labels onto a single unified semantic space, where the images are associated with their corresponding semantic labels.

To formally state the problem, given a training set $\mathcal{D}$ that has $N$ labeled examples, i.e. $\mathcal{D} = \{\boldsymbol{x}_i, y_i\}_{i=1}^N$, where $\boldsymbol{x}_i \in \mathbb{R}^d$ denotes image descriptors and $y_i \in \{1, \ldots, m\}$ are their labels associated with $m$ unique concepts, we want to embed each $\boldsymbol{x}_i$ as $\boldsymbol{z}_i$, and each label $y_i$ as $\boldsymbol{u}_{y_i}$ in the $d_e$-dimensional space, such that the similarity between $\boldsymbol{z}_i$ and $\boldsymbol{u}_{y_i}$, $S(\boldsymbol{z}_i, \boldsymbol{u}_{y_i})$ is maximized.

One way to solve the above problem is to use regression, using $S(\boldsymbol{z}_i, \boldsymbol{u}_{y_i}) = -\|\boldsymbol{z}_i - \boldsymbol{u}_{y_i}\|_2^2$. That is, we estimate the data embedding $\boldsymbol{z}_i$ as $\boldsymbol{z}_i = \boldsymbol{W}\boldsymbol{x}_i$, and minimize their distances to the correct label embeddings $\boldsymbol{u}_{y_i} \in \mathbb{R}^m$ where the dimension for $y_i$ is set to 1 and every other dimension is set to 0:

$$\min_{\boldsymbol{W}} \sum_{c=1}^m \sum_{i=1}^N \|\boldsymbol{W}\boldsymbol{x}_i - \boldsymbol{u}_{y_i}\|_2^2 + \lambda\|\boldsymbol{W}\|_F^2. \tag{1}$$

The above ridge regression will project each instance close to its correct embedding. However, it does not guarantee that the resulting embeddings are well separated. Therefore, most embedding methods for categorization add in discriminative constraints which ensure that the projected instances have higher similarity to their own category embedding than to others. One way to enforce this is to use large-margin constraints on distance: $\|\boldsymbol{W}\boldsymbol{x}_i - \boldsymbol{u}_{y_i}\|_2^2 + 1 \le \|\boldsymbol{W}\boldsymbol{x}_i - \boldsymbol{u}_c\|_2^2 + \xi_{ic}, y_i \ne c$ which can be translated into to the following discriminative loss:

$$\mathcal{L}_C(\boldsymbol{W}, \boldsymbol{U}, \boldsymbol{x}_i, y_i) = \sum_c [1 + \|\boldsymbol{W}\boldsymbol{x}_i - \boldsymbol{u}_{y_i}\|_2^2 - \|\boldsymbol{W}\boldsymbol{x}_i - \boldsymbol{u}_c\|_2^2]_+, \forall c \ne y_i, \tag{2}$$

where $\boldsymbol{U}$ is the columwise concatenation of each label embedding vector, such that $\boldsymbol{u}_j$ denotes $j_{th}$ column of $\boldsymbol{U}$. After replacing the generative loss in the ridge regression formula with the discriminative loss, we get the following discriminative learning problem:

$$\min_{\boldsymbol{W}, \boldsymbol{U}} \sum_i^N \mathcal{L}_C(\boldsymbol{W}, \boldsymbol{U}, \boldsymbol{x}_i, y_i) + \lambda\|\boldsymbol{W}\|_F^2 + \lambda\|\boldsymbol{U}\|_F^2, y_i \in \{1, \ldots, m\}, \tag{3}$$

where $\lambda$ regularizes $\boldsymbol{W}$ and $\boldsymbol{U}$ from shooting to infinity. This is one of the most common objective used for learning discriminative category embeddings for multi-class classification [10, 7], while ranking loss-based [15] models have been also explored for $\mathcal{L}_C$. Bilinear model on a single variable $\boldsymbol{W}$ has been also used in Akata et al. [4], which uses structured labels (attributes) as $\boldsymbol{u}_{y_i}$.

### 3.1 Embedding auxiliary semantic entities.

Now we describe how we embed the supercategories and attributes onto the learned shared space.

**Supercategories.** While our objective is to better categorize *entry* level categories, categories in general can appear at different semantic granularities. For example, a *zebra* could be both an *equus*, and an *odd-toed ungulate*. To learn the embeddings for the supercategories, we map each data instance to be closer to its correct supercategory embedding than to its siblings: $\|\boldsymbol{W}\boldsymbol{x}_i - \boldsymbol{u}_s\|_2^2 + 1 \le \|\boldsymbol{W}\boldsymbol{x}_i - \boldsymbol{u}_c\|_2^2 + \xi_{sc}, \forall s \in \mathcal{P}_{y_i}$ and $c \in \mathcal{S}_s$ where $\mathcal{P}_{y_i}$ denotes the set of superclasses at all levels for class $s$, and $\mathcal{S}_s$ is the set of its siblings. The constraints can be translated into the following loss term:

$$\mathcal{L}_S(\boldsymbol{W}, \boldsymbol{U}, \boldsymbol{x}_i, y_i) = \sum_{s \in \mathcal{P}_{y_i}} \sum_{c \in \mathcal{S}_s} [1 + \|\boldsymbol{W}\boldsymbol{x}_i - \boldsymbol{u}_s\|_2^2 - \|\boldsymbol{W}\boldsymbol{x}_i - \boldsymbol{u}_c\|_2^2]_+. \tag{4}$$

**Attributes.** Attributes can be considered normalized basis vectors for the semantic space, whose combination represents a category. Basically, we want to maximize the correlation between the projected instance that possess the attribute, and its correct attribute embedding, as follows:

$$\mathcal{L}_A(\boldsymbol{W}, \boldsymbol{U}, \boldsymbol{x}_i, y_i) = 1 - \sum_a (\boldsymbol{W}\boldsymbol{x}_i)^\mathsf{T} y_i^a \boldsymbol{u}_a, \|\boldsymbol{u}_a\|^2 \le 1, y_i^a \in \{0, 1\}, \forall a \in \mathcal{A}_{y_i}, \tag{5}$$

where $\mathcal{A}_c$ is the set of all attributes for class $c$ and $\boldsymbol{u}_a$ is an embedding vector for an attribute $a$.

### 3.2 Relationship between the categories, supercategories, and attributes

Simply summing up all previously defined loss functions while adding $\{\boldsymbol{u}_s\}$ and $\{\boldsymbol{u}_a\}$ as additional columns of $\boldsymbol{U}$ will result in a multi-task formulation that implicitly associate the semantic entities, through the shared data embedding $\boldsymbol{W}$. However, we want to further utilize the relationships between the semantic entities, to explicitly impose structural regularization on the semantic embeddings $\boldsymbol{U}$. One simple and intuitive relation is that an object class can be represented as the combination of its parent level category plus a sparse combination of attributes, which translates into the following constraint:

$$\boldsymbol{u}_c = \boldsymbol{u}_p + \boldsymbol{U}^A \boldsymbol{\beta}_c, c \in \mathcal{C}_p, \|\boldsymbol{\beta}_c\|_0 \preceq \gamma_1, \boldsymbol{\beta}_c \succeq 0, \forall c \in \{1, \ldots, \mathsf{C}\}, \tag{6}$$

where $\boldsymbol{U}^A$ is the aggregation of all attribute embeddings $\{\boldsymbol{u}_a\}$, $\mathcal{C}_p$ is the set of children classes for class $p$, $\gamma_1$ is the sparsity parameter, and $\mathsf{C}$ is the number of categories. We require $\boldsymbol{\beta}$ to be non-negative, since it makes more sense and more efficient to describe an object with attributes that it might have, rather than describing it by attributes that it might not have.

We rewrite Eq. 7 into a regularization term as follows, replacing the $\ell_0$-norm constraints with $\ell_1$-norm regularizations for tractable optimization:

$$\mathcal{R}(\boldsymbol{U}, \boldsymbol{B}) = \sum_c^{\mathsf{C}} \|\boldsymbol{u}_c - \boldsymbol{u}_p - \boldsymbol{U}^A \boldsymbol{\beta}_c\|_2^2 + \gamma_2 \|\boldsymbol{\beta}_c + \boldsymbol{\beta}_o\|_2^2,$$
$$c \in \mathcal{C}_p, o \in \mathcal{P}_c \cup \mathcal{S}_c, 0 \preceq \boldsymbol{\beta}_c \preceq \gamma_1, \forall c \in \{1, \ldots, \mathsf{C}\}, \tag{7}$$

where $\boldsymbol{B}$ is the matrix whose $j_{th}$ column vector $\boldsymbol{\beta}_j$ is the reconstruction weight for class $j$, $\mathcal{S}_c$ is the set of all sibling classes for class $c$, and $\gamma_2$ is the parameters to enforce exclusivity.

The exclusive regularization term is used to prevent the semantic reconstruction $\boldsymbol{\beta}_c$ for class $c$ from fitting to the same attributes fitted by its parents and siblings. This is because attributes common across parent and child, and between siblings, are less discriminative. This regularization is especially useful for discrimination between siblings, which belong to the same superclass and only differ by the category-specific modifier. By generating unique semantic decomposition for each class, we can better discriminate between any two categories using a semantic combination of discriminatively learned auxiliary entities.

With the sparsity regularization enforced by $\gamma_1$, the simple sum of the two weights will prevent the two (super)categories from having high weight for a single attribute, which will let each category embedding to fit to exclusive attribute set. This, in fact, is the exclusive lasso regularizer introduced in [16], except for the nonnegativity constraint on $\boldsymbol{\beta}_c$, which makes the problem easier to solve.

### 3.3 Unified semantic embeddings for object categorization

After augmenting the categorization objective in Eq. 3 with the superclass and attributes loss and the sparse-coding based regularization in Eq. 7, we obtain the following multitask learning formulation that jointly learns all the semantic entities along with the sparse-coding based regularization:

$$\min_{\boldsymbol{W},\boldsymbol{U},\boldsymbol{B}} \sum_{i=1}^{N} \mathcal{L}_C(\boldsymbol{W},\boldsymbol{U},\boldsymbol{x}_i,y_i) + \mu_1 \left(\mathcal{L}_S(\boldsymbol{W},\boldsymbol{U},\boldsymbol{x}_i,y_i) + \mathcal{L}_A(\boldsymbol{W},\boldsymbol{U},\boldsymbol{x}_i,y_i)\right) + \mu_2 \mathcal{R}(\boldsymbol{U},\boldsymbol{B}); \tag{8}$$

$$\|\boldsymbol{w}_j\|_2^2 \leq \lambda, \|\boldsymbol{u}_k\|_2^2 \leq \lambda, 0 \preceq \boldsymbol{\beta}_c \preceq \gamma_1 \forall j \in \{1,\ldots,d\}, \forall k \in \{1,\ldots,m\}, \forall c \in \{1,\ldots,\mathsf{C}+\mathsf{S}\},$$

where $\mathsf{S}$ is the number of supercategories, $\boldsymbol{w}_j$ is $\boldsymbol{W}$'s $j_{th}$ column, and $\mu_1$ and $\mu_2$ are parameters to balance between the main and auxiliary tasks, and discriminative and generative objective.

Eq. 8 could be also used for knowledge transfer when learning a model for a novel set of categories, by replacing $\boldsymbol{U}^A$ in $\mathcal{R}(\boldsymbol{U},\boldsymbol{B})$ with $\boldsymbol{U}^{\mathcal{S}}$, learned on class set $\mathcal{S}$ to transfer the knowledge from.

### 3.4 Numerical optimization

Eq. 8 is not jointly convex in all variables, and has both discriminative and generative terms. This problem is similar to the problem in [14], where the objective is to learn the dictionary, sparse coefficients, and classifier parameters together, and can be optimized using a similar alternating optimization, while each subproblem differs. We first describe how we optimize for each variable.

**Learning of $\boldsymbol{W}$ and $\boldsymbol{U}$.** The optimization of both embedding models are similar, except for the reconstructive regularization on $\boldsymbol{U}$. and the main bottleneck lies in the minimization of the $\mathcal{O}(Nm)$ large-margin losses. Since the losses are non-differentiable, we solve the problems using stochastic subgradient method. Specifically, we implement the proximal gradient algorithm in [17], handling the $\ell$-2 norm constraints with proximal operators.

**Learning $\boldsymbol{B}$.** This is similar to the sparse coding problem, but simpler. We use projected gradient method, where at each iteration $t$, we project the solution of the objective $\boldsymbol{\beta}_c^{t+\frac{1}{2}}$ for category $c$ to $\ell$-1 norm ball and nonnegative orthant, to obtain $\boldsymbol{\beta}_c^t$ that satisfies the constraints.

**Alternating optimization.** We decompose Eq. 8 to two convex problems: 1) Optimization of the data embedding $\boldsymbol{W}$ and approximation parameter $\boldsymbol{B}$ (Since the two variable do not have direct link between them) , and 2) Optimization of the category embedding $\boldsymbol{U}$. We alternate the process of optimizing each of the convex problems while fixing the remaining variables, until the convergence criterion [1] is met, or the maximum number of iteration is reached.

**Run-time complexity.** *Training:* Optimization of $\boldsymbol{W}$ and $\boldsymbol{U}$ using proximal stochastic gradient [17], have time complexities of $O(d^e d(k+1))$ and $O(d^e(dk+C))$ respectively. Both terms are dominated by the gradient computation for $k(k \ll N)$ sampled constraints, that is $O(d^e dk)$. Outer loop for alternation converges within 5-10 iterations depending on $\epsilon$. *Test:* Test time complexity is exactly the same as in LME, which is $O(d_e(C+d))$.

## 4 Results

We validate our method for multiclass categorization performance on two different datasets generated from a public image collection, and also test for knowledge transfer on few-shot learning.

### 4.1 Datasets

We use Animals with Attributes dataset [1], which consists of $30,475$ images of 50 animal classes, with $85$ class-level attributes [2]. We use the Wordnet hierarchy to generate supercategories. Since

there is no fixed training/test split, we use {30,30,30} random split for training/validation/test. We generate the following two datasets using the provided features. **1) AWA-PCA:** We compose a 300-dimensional feature vectors by performing PCA on each of 6 types of features provided, including SIFT, rgSIFT, SURF, HoG, LSS, and CQ to have 50 dimensions per each feature type, and concatenating them. **2) AWA-DeCAF:** For the second dataset, we use the provided 4096-D DeCAF features [18] obtained from the layer just before the output layer of a deep convolutional neural network.

## 4.2 Baselines

We compare our proposed method against multiple existing embedding-based categorization approaches, that either do not use any semantic information, or use semantic information but do not explicitly embed semantic entities. For non-semantics baselines, we use the following: **1)Ridge Regression:** A linear regression with $\ell$-2 norm (Eq. 1). **2) NCM:** Nearest mean classifier from [11], which uses the class mean as category embeddings ($\boldsymbol{u}_c = \boldsymbol{x}_c^\mu$). We use the code provided by the authors[3]. **3) LME:** A base large-margin embedding (Eq. 3) solved using alternating optimization.

For implicit semantic baselines, we consider two different methods. **4) LMTE:** Our implementation of the Weinberger et al. [7], which enforces the semantic similarity between class embeddings as distance constraints [7], where $\boldsymbol{U}$ is regularized to preserve the pairwise class similarities from a given taxonomy. **5-7) ALE, HLE, AHLE:** Our implementation of the attribute label embedding in Akata et al. [4], which encodes the semantic information by representing each class with structured labels that indicate the class' association with superclasses and attributes. We implement variants that use attributes (ALE), leaf level + superclass labels (HLE), and both (AHLE) labels.

For our models, we implement multiple variants to analyze the impact of each semantic entity and the proposed regularization. **1) LME-MTL-S:** The multitask semantic embedding model learned with supercategories. **2) LME-MTL-A:** The multitask embedding model learned with attributes. **3) USE-No Reg.:** The unified semantic embedding model learned using both attributes and supercategories, without semantic regularization. **4) USE-Reg:** USE with the sparse coding regularization.

For parameters, the projection dimension $d_e = 50$ for all our models. [4] For other parameters, we find the optimal value by cross-validation on the validation set. We set $\mu_1 = 1$ that balances the main and auxiliary task equally, and search for $\mu_2$ for discriminative/generative tradeoff, in the range of $\{0.01, 0.1, 0.2 \ldots, 1, 10\}$, and set $\ell$-2 norm regularization parameter $\lambda = 1$. For sparsity parameter $\gamma_1$, we set it to select on average several (3 or 4) attributes per class, and for disjoint parameter $\gamma_2$, we use $10\gamma_1$, without tuning for performance.

| | | Flat hit @ k (%) | | | Hierarchical precision @ k (%) | |
|---|---|---|---|---|---|---|
| | Method | 1 | 2 | 5 | 2 | 5 |
| No semantics | Ridge Regression | $19.31 \pm 1.15$ | $28.34 \pm 1.53$ | $44.17 \pm 2.33$ | $28.95 \pm 0.54$ | $39.39 \pm 0.17$ |
| | NCM [11] | $18.93 \pm 1.71$ | $29.75 \pm 0.92$ | $47.33 \pm 1.60$ | $30.81 \pm 0.53$ | $43.43 \pm 0.53$ |
| | LME | $19.87 \pm 1.56$ | $30.47 \pm 1.56$ | $48.07 \pm 1.06$ | $30.98 \pm 0.62$ | $42.63 \pm 0.56$ |
| Implicit semantics | LMTE [7] | $20.76 \pm 1.64$ | $30.71 \pm 1.35$ | $47.76 \pm 2.25$ | $31.05 \pm 0.71$ | $43.13 \pm 0.29$ |
| | ALE [4] | $15.72 \pm 1.14$ | $25.63 \pm 1.44$ | $43.42 \pm 1.67$ | $29.26 \pm 0.50$ | $43.71 \pm 0.34$ |
| | HLE [4] | $17.09 \pm 1.09$ | $27.52 \pm 1.20$ | $45.49 \pm 0.61$ | $30.51 \pm 0.48$ | $44.76 \pm 0.20$ |
| | AHLE [4] | $16.65 \pm 0.47$ | $26.55 \pm 0.77$ | $43.05 \pm 1.22$ | $29.49 \pm 0.89$ | $43.41 \pm 0.65$ |
| Explicit semantics | LME-MTL-S | $20.77 \pm 1.41$ | $32.09 \pm 1.67$ | $50.94 \pm 1.21$ | $33.71 \pm 0.94$ | $45.73 \pm 0.71$ |
| | LME-MTL-A | $20.65 \pm 0.83$ | $31.51 \pm 0.72$ | $49.40 \pm 0.62$ | $31.69 \pm 0.49$ | $43.47 \pm 0.23$ |
| USE | USE-No Reg. | $21.07 \pm 1.53$ | $31.59 \pm 1.57$ | $50.11 \pm 1.51$ | $\mathbf{33.67 \pm 0.55}$ | $45.41 \pm 0.43$ |
| | USE-Reg. | $\mathbf{21.64 \pm 1.02}$ | $\mathbf{32.69 \pm 0.83}$ | $\mathbf{52.04 \pm 1.02}$ | $33.37 \pm 0.74$ | $\mathbf{47.17 \pm 0.91}$ |

Table 1: Multiclass classification performance on **AWA-PCA** dataset (300-D PCA features).

## 4.3 Multiclass categorization

We first evaluate the suggested multitask learning framework for categorization performance. We report the average classification performance and standard error over 5 random training/test splits in Table 1 and 2, using both flat hit@k, which is the accuracy for the top-k predictions made, and hierarchical precision@k from [12], which is a precision the given label is correct at $k$, at all levels.

Non-semantic baselines, ridge regression and NCM, were outperformed by our most basic LME model. For implicit semantic baselines, ALE-variants underperformed even the ridge regression

| | | Flat hit @ k (%) | | | Hierarchical precision @ k (%) | |
|---|---|---|---|---|---|---|
| | Method | 1 | 2 | 5 | 2 | 5 |
| No semantics | Ridge Regression | 38.39 ± 1.48 | 48.61 ± 1.29 | 62.12 ± 1.20 | 38.51 ± 0.61 | 41.73 ± 0.54 |
| | NCM [11] | 43.49 ± 1.23 | 57.45 ± 0.91 | 75.48 ± 0.58 | 45.25 ± 0.52 | 50.32 ± 0.47 |
| | LME | 44.76 ± 1.77 | 58.08 ± 2.05 | 75.11 ± 1.48 | 44.84 ± 0.98 | 49.87 ± 0.39 |
| Implicit semantics | LMTE [7] | 38.92 ± 1.12 | 49.97 ± 1.16 | 63.35 ± 1.38 | 38.67 ± 0.46 | 41.72 ± 0.45 |
| | ALE [4] | 36.40 ± 1.03 | 50.43 ± 1.92 | 70.25 ± 1.97 | 42.52 ± 1.17 | 52.46 ± 0.37 |
| | HLE [4] | 33.56 ± 1.64 | 45.93 ± 2.56 | 64.66 ± 1.77 | 46.11 ± 2.65 | **56.79 ± 2.05** |
| | AHLE [4] | 38.01 ± 1.69 | 52.07 ± 1.19 | 71.53 ± 1.41 | 44.43 ± 0.66 | 54.39 ± 0.55 |
| Explicit semantics | LME-MTL-S | 45.03 ± 1.32 | 57.73 ± 1.75 | 74.43 ± 1.26 | 46.05 ± 0.89 | 51.08 ± 0.36 |
| | LME-MTL-A | 45.55 ± 1.71 | 58.60 ± 1.76 | 74.97 ± 1.15 | 44.23 ± 0.95 | 48.52 ± 0.29 |
| USE | USE-No Reg. | 45.93 ± 1.76 | 59.37 ± 1.32 | 74.97 ± 1.15 | 47.13 ± 0.62 | 51.04 ± 0.46 |
| | USE-Reg. | **46.42 ± 1.33** | **59.54 ± 0.73** | **76.62 ± 1.45** | **47.39 ± 0.82** | 53.35 ± 0.30 |

Table 2: Multiclass classification performance on **AWA-DeCAF** dataset (4096-D DeCAF features).

baseline with regard to the top-1 classification accuracy [5], while they improve upon the top-2 recognition accuracy and hierarchical precision. This shows that hard-encoding structures in the label space do not necessarily improve the discrimination performance, while it helps to learn a more semantic space. LMTE makes substantial improvement on 300-D features, but not on DeCAF features.

Explicit embedding of semantic entities using our method improved both the top-1 accuracy and the hierarchical precision, with USE variants achieving the best performance in both. Specifically, adding superclass embeddings as auxiliary entities improves the hierarchical precision, while using attributes improves the flat top-k classification accuracy. USE-Reg, especially, made substantial improvements on flat hit and hierarchical precision @ 5, which shows the proposed regularization's effectiveness in learning a semantic space that also discriminates well.

| Category | Ground-truth attributes | Supercategory + learned attributes |
|---|---|---|
| 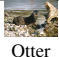 Otter | An animal that swims, fish, water, new world, small, flippers, furry, black, brown, tail, ... | A musteline mammal that is quadrapedal, flippers, furry, ocean |
|  Skunk | An animal that is smelly, black, stripes, white, tail, furry, ground, quadrapedal, new world, walks, ... | A musteline mammal that has stripes |
|  Deer | An animal that is brown, fast, horns, grazer, forest, quadrapedal, vegetation, timid, hooves, walks, ... | A deer that has spots, nestspot, longneck, yellow, hooves |
|  Moose | An animal that has horns, brown, big, quadrapedal, new world, vegetation, grazer, hooves, strong, ground,... | A deer that is arctic, <span style="color:red">stripes</span>, black |
| Equine | N/A | An odd-toed ungulate, that is lean and active |
| Primate | N/A | An animal, that has hands and bipedal |

Table 3: Semantic description generated using ground truth attributes labels and learned semantic decomposition of each categorys. For ground truth labels, we show top-10 ranked by their human-ranked relevance. For our method, we rank the attributes by their learned weights. Incorrect attributes are colored in red.

### 4.3.1 Qualitative analysis

Besides learning a space that is both discriminative and generalizes well, our method's main advantage, over existing methods, is its ability to generate compact, semantic descriptions for each category it has learned. This is a great caveat, since in most models, including the state-of-the art deep convolutional networks, humans cannot understand what has been learned; by generating human-understandable explanation, our model can *communicate* with the human, allowing understanding of the rationale behind the categorization decisions, and to possibly allow feedback for correction.

To show the effectiveness of using supercategory+attributes in the description, we report the learned reconstruction for our model, compared against the description generated by its ground-truth attributes in Table 3. The results show that our method generates compact description of each category, focusing on its *discriminative* attributes. For example, our method select attributes such as *flippers* for otter, and *stripes* for skunk, instead of attributes common and nondescriminative such as *tail*. Note that some attributes that are ranked less relevant by humans were selected for their discriminativity, e.g., *yellow* for dear and *black* for moose, both of which human annotators regarded

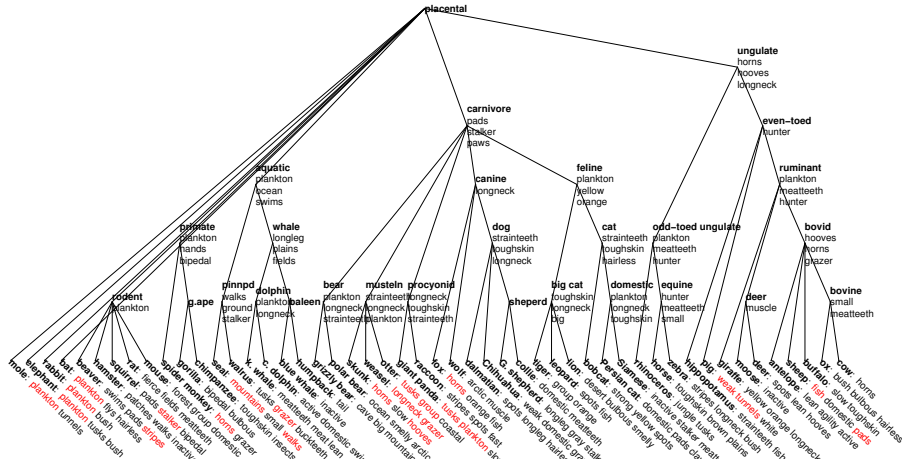

Figure 2: Learned discriminative attributes association on the AWA-PCA dataset. Incorrect attributes are colored in red.

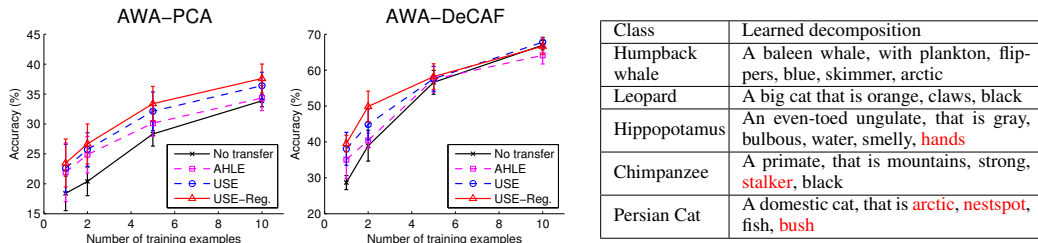

| Class | Learned decomposition |
|---|---|
| Humpback whale | A baleen whale, with plankton, flippers, blue, skimmer, arctic |
| Leopard | A big cat that is orange, claws, black |
| Hippopotamus | An even-toed ungulate, that is gray, bulbous, water, smelly, hands |
| Chimpanzee | A primate, that is mountains, strong, stalker, black |
| Persian Cat | A domestic cat, that is arctic, nestspot, fish, bush |

Figure 3: Few-shot experiment result on the AWA dataset, and generated semantic decompositions.

as *brown*. Further, our method selects discriminative attributes for each supercategory, while there is no provided attribute label for supercategories.

Figure 2 shows the discriminative attributes disjointly selected at each node on the class hierarchy. We observe that coarser grained categories fit to attributes that are common throughout all its children (e.g. *pads*, *stalker* and *paws* for carnivore), while the finer grained categories fit to attributes that help for finer-grained distinctions (e.g. *orange* for tiger, *spots* for leopard, and *desert* for lion).

## 4.4 One-shot/Few-shot learning

Our method is expected to be especially useful for few-shot learning, by generating a richer description than existing methods, that approximate the new input category using either trained categories or attributes. For this experiment, we divide the $50$ categories into predefined $40/10$ training/test split, and compare with the knowledge transfer using AHLE. We assume that no attribute label is provided for test set. For AHLE, and USE, we regularize the learning of $W$ with $W^\mathcal{S}$ learned on training class set $\mathcal{S}$ by adding $\lambda_2 \|W - W^\mathcal{S}\|_2^2$, to LME (Eq. 3). For USE-Reg we use the reconstructive regularizer to regularize the model to generate semantic decomposition using $U^\mathcal{S}$.

Figure 3 shows the result, and the learned semantic decomposition of each novel category. While all methods make improvements over the no-transfer baseline, USE-Reg achieves the most improvement, improving two-shot result on AWA-DeCAF from 38.93% to 49.87%, where USE comes in second with 44.87%. Most learned reconstructions look reasonable, and fit to discriminative traits that help to discriminate between the test classes, which in this case are colors; *orange* for leopard, *gray* for hippopotamus, *blue* for humpback whale, and *arctic* (white) for Persian cat.

## 5 Conclusion

We propose a unified semantic space model that learns a discriminative space for object categorization, with the help of auxiliary semantic entities such as supercategories and attributes. The auxiliary entities aid object categorization both indirectly, by sharing a common data embedding, and directly, by a sparse-coding based regularizer that enforces the category to be generated by its supercategory + a sparse combination of attributes. Our USE model improves both the flat-hit accuracy and hierarchical precision on the AWA dataset, and also generates semantically meaningful decomposition of categories, that provides human-interpretable rationale.

## Footnotes

[1] $\|\boldsymbol{W}^{t+1} - \boldsymbol{W}^t\|_2 + \|\boldsymbol{U}^{t+1} - \boldsymbol{U}^t\|_2 + \|\boldsymbol{B}^{t+1} - \boldsymbol{B}^t\|_2 < \epsilon$

[2] Attributes are defined on color (black, orange), texture (stripes, spots), parts (longneck, hooves), and other high-level behavioral properties (slow, hibernate, domestic) of the animals

[3]http://staff.science.uva.nl/~tmensink/code.php

[4]Except for ALE variants where $d_e$=m, the number of semantic entities.

[5]We did extensive parameter search for the ALE variants.

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
