[Reviews · NeurIPS 2014]

Submitted by Assigned_Reviewer_22

The paper presents a multi-task learning based technique for jointly accommodating information conveyed by mid-level attributes and category semantics. A sparse coding approach is utilized to infer discriminative information to identify categories while permitting generative modeling to reason out new categories. The method is evaluated on the Animals with attributes dataset and shows improved classification performance, with an emphasis on one/few-shot learning circumstances.

The paper, as mentioned in the manuscript, is a culmination of different existing ideas pertaining to semantic models, discriminative embedding, multitask learning and sparse learning. While the paper does argue how they are different, in my opinion there isn't much contribution in terms of a new approach/algorithm.

Moreover, the paper seems to overly use/rely on recent "terminologies" in the name of attributes/semantics, while in crux all its doing is taking different pieces of information about the entities (object categories, here) one is interested in. While one can argue, for instance, attributes are generic while category specific information is not, the approach is more like an hierarchical one where such information can be integrated. In that spirit, existing deep learning methods do the same thing without having to handcode these pieces of information. Also of relevance is the coarse-to-fine strategies that vision community has looked at for many years. Besides dealing with these features, the claim of jointly performing discriminative and generative modeling dates back atleast three decades in the pattern recognition literature eg [Ref1-4]. And in doing do, the paper resorts to existing sparse coding schemes - while it argues [14] learns dictionary in unsupervised way, there are efforts that learn dictionaries discriminatively (eg. [Ref5-6]). Thus there is not much new/interesting contribution in the paper.

References:
[Ref1] P. A. Devijver and J. Kittler, Pattern Recognition: A Statistical Approach (Prentice-Hall International, Englewood Cliffs, NJ, 1980).
[Ref2] K. Fukunaga, Introduction to Statistical Pattern Recognition, 2nd Ed. (Academic Press, New York, 1990.
[Ref3] R. Schalkoff, Pattern Recognition: Statistical, Structural and Neural Approaches (John Wiley & Sons, New York, 1992)
[Ref4] Lasserre, Julia A., Christopher M. Bishop, and Thomas P. Minka. "Principled hybrids of generative and discriminative models." Computer Vision and Pattern Recognition, 2006 IEEE Computer Society Conference on. Vol. 1. IEEE, 2006.
[Ref5] Jiang, Zhuolin, Zhe Lin, and Larry S. Davis. "Learning a discriminative dictionary for sparse coding via label consistent K-SVD." Computer Vision and Pattern Recognition (CVPR), 2011 IEEE Conference on. IEEE, 2011.
[Ref6] Yang, Meng, D. Zhang, and Xiangchu Feng. "Fisher discrimination dictionary learning for sparse representation." Computer Vision (ICCV), 2011 IEEE International Conference on. IEEE, 2011.
Summary: A system paper that pieces together several existing concepts, while what it really does is covered by several aspects of existing literature that are not referred to (some are given in references). Hence the paper does not have any significant contribution to support its acceptance.

Submitted by Assigned_Reviewer_41

This work presents a method for modeling object categories, supercategories, and attributes in a shared space. This is made possible by an appropriately labeled dataset: Animal with Attributes. The main motivation is that the category-supercategory and category-attribute relationships provide generative regularization for the category-only discriminative learning.

[strength] The work is well motivated in the introduction and related work. I recommend some mention of recent deep methods in the "sparse coding" subsection.

[suggestion] Figure 1 could more efficiently use available space by type-setting text to the right of the figure.

[strength] The approach is clearly laid out component by component, with clear mathematical notation, appropriate level of detail, and a sufficient explanation of the numerical implementation.

[strength] Evaluation considers a fair set of baselines, and uses two different underlying features to demonstrate that the method is not overfit to a particular feature space. It is unfrotunate that only the AWA dataset is considered, but perhaps there are no other possibilities for this involved task.

[weakness] Little discussion of the computational concerns of this method. The optimization looks tricky, and I would appreciate knowing details of the training and test runtimes.
Summary: The proposed unified semantic embedding model is well motivated, clearly and sequentially formulated, and properly evaluated -- demonstrating superiority -- on one super-labeled dataset. The paper would further benefit through discussing computational concerns.

Submitted by Assigned_Reviewer_42

The paper introduces methodology to learn a semantic space for object categorization, where semantic entities like supercategories and attributes are used to constrain the resulting (discriminative) embedding. The main idea is to (sparsely) represent a category by means of its super-category + a combination of attributes (tiger=striped feline). An advantage of the method is the capacity to generate compact semantic descriptions for the learnt categories. The paper is clearly presented, the motivation is sound and the good results well-emphasize the superiority of the propose techniques. While some of the principles of designing cost functions for discriminative embeddings with good inter-class separation have been described elsewhere (e.g. [7], [14]), the authors here present novel ways to instantiate such ideas in order to connect categories, supercategories and attributes.

- Other Comments

The introduction is a bit confusing. Too many elements (concepts) float around without a clear buildup. On lines 077-079 it is unclear what the generative and the discriminative objectives are until later. The combination of precise terminology (e.g. generative/discriminative) and imprecise anchoring in the particular context makes the text more difficult to follow.

Line 145: S(z_i,…) not S(z,…)

The models are non-convex. How is the initialization performed?
Summary: A compact semantic space model that learns a discriminative space for object categorization by leveraging constraints from supercategories and attributes. Improved results on AWA dataset with the additional advantage of a model that can generate a human-interpretable decomposition of categories.

Submitted by Meta_Reviewer_10

Additional observations by AC.

There are now quite a few similar models in the semantic embedding
arena so it does get confusing. Consider working top down to help the
reader along, i.e. outlining the things covered, then (8), then
explaining each piece in detail.

(4) fits with (2) but it is perhaps overkill. Would much be lost if
supercategory u's were simply modelled as convex combinations of their
children? Or even as subspaces (simplices) not points? (The simplex
viewpoint might allow fine-grainedness to be modelled more naturally,
with only very specific categories becoming truly point-like).

It might be more natural to view attributes as linear forms not
point-vectors (i.e. as dimensions of semantic space or "linear
detectors" in input space). So, e.g., cutting the supercategory
simplex by the attribute linear forms or half spaces would give a
smaller simplex representing the subcategory, ad infinitum.
Summary: An interesting paper on combining attributes and hierarchies in semantic embedding.
Author Feedback
Author rebuttal: We thank all reviewers for their effort and helpful feedback. We are pleased to see that R2 and R3 appreciate the method’s novelty in drawing the relationships between attributes and supercategories, and generating semantic description.

== R1 (R22) ==
R1 is concerned with method’s originality and seems to perceive it as a system that pieces together existing concepts. This is not the case. We believe confusion stems from misunderstanding of core claims and advantages of our method.

The main novelty of our approach is the idea to formulate semantic relations of the form ‘object category = supercategory + sparse combination of attributes’ into structural constraints to regularize the learning of the discriminative space (Fig 1, L081-087). This is acknowledged by both R2 and R3, and has not been addressed in any past literature (including list of references provided). We present a ‘single’ unified learning framework (Eq.8) with a novel structural regularization based on semantic relations (Eq.6,7). This core idea and resulting formulation, leads to the main advantage of our method which is its ability to generate compact, semantic descriptions for the learned categories (L029-030, R3’s summary). R1’s summary misses out this core idea and advantage.

1) what all attributes / semantics are doing is taking different pieces of information…
=> Our method focus on how to translate the ‘semantic relations’ into structural constraints in the regularized learning formulation. Such semantic-constraint associations are not something that is given, and need to be modeled (Eq 6,7). We discuss such modeling efforts in Answer 5.

2) Existing deep learning method can do the same without handcoding information
=> Our method provides the semantic rationale behind the predictions, e.g., ‘tiger = striped feline’, by generating compact semantic descriptions of the learned object categories. Such semantic rationale is not provided by the deep-learning methods. Moreover, the two methods are complementary. The deep learning models can be used to learn underlying image representation that feeds into our model; we show this by experimenting on DeCAF features (Table 2), which are deep representations.

3) The claim of jointly performing discriminative and generative modeling dates back...
=> We do not make the claim of inventing joint discriminative and generative modeling, our core claim is stated above. We are aware of the literature and even clearly mention that [14] has the similar combined objective (L125-131, 241-244). This, however, does not invalidate our main claim and novelty as stated above.

4) The paper resort to existing sparse coding schemes… there are efforts that learn dictionaries discriminatively
=> Our method is a discriminative embedding method with structural regularization (see the summary from R2), which has clearly different learning objective (Eq.3,4,5,8) from that of mentioned dictionary learning methods.

= Differences in the combined objective
- Our discriminative objective is a large-margin embedding [10,11,7,4,12,9], that learns a space to classify embedded data instances by their distances to category prototypes.
- The sparse-coding-based term ‘regularizes’ the categorization learning objective with the semantic constraints, while dictionary learning approaches in [14], [Ref5], and [Ref6] learn ‘mid-level representations’. Therefore, in our method, there is no dependency between discriminative and generative objective as in [14,Ref5,REf6].

= Differences in the sparse-coding term
- Our sparse-coding term allows hierarchical relations among categories with category = supercategory + attributes constraints for compact decomposition (Eq.6), which results in the dependencies between arbitrary level of dictionary atoms (See Fig 1.)
- We enforce exclusivity between different models, which enables us to learn exclusive semantic description for a category that is different from its parent and siblings (Eq.7).

5) Missing references
=> In related work, we mainly focused on semantic and discriminative embedding methods that are directly related, due to the page limit and to ensure coherence. Suggested [Ref1-4] are more distantly related (also see answer to Q4). We discuss relationship of our method to [Ref5,6] above and will add this discussion to the related work section.

== R2 (R41) ==
1) Some mention of recent deep methods
=> Assuming space allows, we will add a brief discussion per suggestion.

2) Training and test runtimes.
=> We will add the following in the revision, if paper is accepted:
- Training time: W and U are optimized using the proximal stochastic gradient [17], with time complexities of O(d^e x d x (k + 1)) and O(d^e x (d x k + C)) respectively. Both terms are dominated by the gradient computation for k (k< < N) sampled constraints, that is O(d^e x d x k). Outer loop for alternation converges within 5-10 iterations depending on e (L268). On an i7 machine with 32GB memory, the avg. training time for 1500 training instances with 4096-D features is 8.2 hrs (33 min for 300-D).
- Test time: Exactly the same as LME, which is O(d_e x (C+d)). On average, it takes 0.06 sec to process 1500 test instances with 4096-D features (0.01 sec for 300-D).

== R3 (R42) ==
1) Unclear text about discriminative and generative objective (L077-079)
=> We state the discriminative objective in (L045-046) and the generative objective in (L073-075), but admit that presentation is somewhat confusing and will edit the intro to clarify.

2) Initialization
=> We initialize U by setting each category embedding u_c as the PCA-projected sample mean for each class c (in d_e target dimensional space). W is initialized with the solution of ridge regression using this initial U. We also tested random initialization with Gaussian noise for each column vector, but found no significant difference in performance with sufficient number of alternations (>5).